# A Brief Chronicle of Antibody Research and Technological Advances

**DOI:** 10.3390/antib13040090

**Published:** 2024-11-11

**Authors:** Kazutaka Araki, Ryota Maeda

**Affiliations:** 1AIST-UTokyo Advanced Operando-Measurement Technology Open Innovation Laboratory (Operando-Oil), National Institute of Advanced Industrial Science and Technology (AIST), 6-2-3 Kashiwanoha, Chiba 277-0882, Japan; 2COGNANO Inc., 64-101 Kamitakano Higashiyama, Sakyo-ku, Kyoto 601-1255, Japan; maeda@cognano.co.jp

**Keywords:** antibody, immunoglobulin, immunology

## Abstract

This review briefly traces the historical development of antibody research and related technologies. The path from early perceptions of immunity to the emergence of modern immunotherapy has been marked by pivotal discoveries and technological advances. Early insights into immunity led to the development of vaccination and serotherapy. The elucidation of antibody structure and function paved the way for monoclonal antibody technology and its application in diagnosis and therapy. Breakthroughs in genetic engineering have enabled the production of humanized antibodies and the advances in Fc engineering, thereby increasing therapeutic efficacy. The discovery of immune checkpoints and cytokines revolutionized the treatment of cancer and autoimmune diseases. The field continues to evolve rapidly with the advent of antibody–drug conjugates, bispecific antibodies, and CAR T-cell therapies. As we face global health challenges, antibody research remains at the forefront of medical innovation and offers promising solutions for the future.

## 1. Introduction

This review aims to provide a brief historical overview of the significant scientific discoveries and technological advances related to antibodies rather than encompassing this field [1,2]. Antibodies are invaluable in research and increasingly practical in clinical applications due to their high specificity and affinity in binding targets [3,4]. The Y-shaped structure of antibodies serves as a classic symbol in science (Figure 1) [5]. Astrid Fagraeus identified plasma cells as the source of antibodies [6]. Macfarlane Burnet proposed the clonal selection theory, which transformed our understanding of the immune system by explaining how specific antibodies are developed to combat infections [7]. Max Cooper discovered the tissue responsible for producing B cells, precursors of plasma cells [8]. Georges Köhler and César Milstein’s 1975 paper on monoclonal antibodies revolutionized antibody research and therapeutic development [9]. Susumu Tonegawa’s 1983 paper on gene recombination in B cells explained how antibodies could target various antigens [10]. Greg Winter’s research on antibody humanization was critical to clinical applications [11]. Before reviewing these iconic achievements, let us look at the early beginnings of immunology.

## 2. The Antibody Research and Related Technologies

### 2.1. Early Beginnings: Inoculation and Vaccination (18th–19th Century) 

The roots of immunology can be traced back to ancient civilizations, where observations of immunity to disease were recorded [1]. In the 18th century, more systematic approaches to harnessing the power of immunity emerged in Europe, building on practices already established in parts of Asia [13]. While Lady Mary Wortley Montagu is often credited with introducing smallpox inoculation (variolation) to England, Emanuel Timoni [14] and Giacomo Pylarini [15] made earlier contributions (Figure 2). In 1714, their texts were published together in the same volume, which helped to amplify their impact in Europe despite their individual contributions.

Lady Mary played an important role in popularizing vaccination in England after observing it in Turkey around 1717. She boldly inoculated her son in Constantinople in 1718 and inoculated her daughter in 1721, the first such procedure on English soil [27]. Although it reduced mortality compared to natural infection, it carried risks of severe illness, death, and the spread of disease.

A breakthrough came in 1796 with Edward Jenner’s smallpox vaccination experiments [28], later published in 1798 [16]. His famous experiment involved inoculating an 8-year-old boy, James Phipps, with material from a cowpox lesion on the hand of Sarah Nelms, a milkmaid. When later exposed to smallpox, Phipps showed no signs of infection. Jenner’s work introduced the concept of vaccination (from the Latin “vacca” for cow). In 1840, the British government banned variolation and provided free cowpox vaccination, an important step in public health policy [29]. One hundred and forty years later, on 8 May 1980, the 33rd World Health Assembly formally declared: “The world and all its peoples have achieved freedom from smallpox” [30].

It is worth noting that the development of medical science during this period was not limited to smallpox prevention. In 1848, Henry Bence Jones published a paper describing a new substance found in a patient’s urine with mollities ossium (softening of the bones) [17]. This discovery, later known as the Bence Jones protein or immunoglobulin light chain, became a diagnostic marker for multiple myeloma [31].

### 2.2. The Birth of Serum Therapy (Late 19th Century) 

In 1888, George Nuttall demonstrated that defibrinated blood had significant bactericidal activity against anthrax bacilli, which lost its activity when heated to 55 °C [32]. This discovery pioneered humoral immunity research. In 1890, Emil von Behring made innovative discoveries regarding diphtheria immunity in animals, demonstrating that the pretreatment of animals with hydrogen peroxide could confer varying degrees of immunity to diphtheria (Figure 2) [20]. Later that year, Behring collaborated with Shibasaburo Kitasato on a seminal paper on animal tetanus immunity [33]. Their experiments showed that blood serum from animals with acquired immunity could be used to both cure and prevent infections in other animals. This discovery led to the development of serum therapy, i.e., serotherapy, and introduced the basic concept of antibodies to immunology. The practical application of these discoveries came quickly. In 1891, the first human trials of diphtheria antitoxin in Berlin, Germany, showed promising results, leading to its rapid adoption in Europe and North America [34]. That same year, antitoxins were documented as “globulins” that could precipitate when serum was mixed with magnesium sulfate in an experiment by Guido Tizzoni and Giuseppina Cattani [35].

Antitoxin production was soon industrialized, and the Institut Pasteur began large-scale production in 1894 [36]. This industrialization marked the beginning of biological therapeutics and established the groundwork for the modern pharmaceutical industry. Behring was awarded the first Nobel Prize in Physiology or Medicine in 1901. The Nobel Committee’s decision was influenced by the severe impact of diphtheria at the time and Behring’s contributions to its treatment.

### 2.3. Foundations of Modern Immunology (19th–20th Century) 

The concept of cellular immunity began to emerge in the mid-19th century. In 1862, Ernst Haeckel observed that hemolymph cells in mollusks demonstrated phagocytic behavior (Figure 2) [18]. The 1884 work by Ilya Ilyich Mechnikov, also spelled Élie Metchnikoff, on phagocytosis laid the foundation for understanding innate immunity [19]. He inserted a rose thorn into a starfish larva and observed phagocytic cells rapidly migrating to and clustering around the thorn. This observation led to the proposal that specialized cells called phagocytes were responsible for engulfing and destroying foreign particles and microorganisms [37].

In parallel, significant advancements were made in humoral immunology. In 1895, Jules Bordet discovered the complement system [38]. He demonstrated that bacteriolysis required two components: a specific antibody and a heat-labile factor present in all animals, which he termed “alexin”. This factor is now known as complement, and its mechanism is referred to as Complement-Dependent Cytotoxicity (CDC).

In the same era, Paul Ehrlich played a vital role in humoral immunology, discovering mast cells with aniline staining [39], coining the term “antibody” (Antikörper in German) in 1891 [21,40], and proposing the “side chain theory” in 1897 [22]. This side chain theory suggests that cells express different receptors, or side chains, which can be shed into the blood as antitoxins or antibodies. When a toxin binds to a cell’s side chain, the cell overproduces these chains, which are then released to neutralize toxins [41,42]. The released receptors, now known as antibodies, can neutralize the toxins by binding to them, providing the first insights into antibody–antigen interactions. Ehrlich also described how maternal antibodies can transfer to offspring and protect them from infections early in life [43]. Incidentally, in 1899, Ladislav Deutsch coined the term “antigen” to describe bacterial products that may become antibodies [23].

Ehrlich’s work was supported by Karl Landsteiner’s discovery of blood groups in 1900 [24]. This discovery demonstrated the presence of specific antigens on red blood cells and explained the phenomenon of blood compatibility and incompatibility. The importance of this field was recognized in 1908 when Ehrlich and Mechnikov shared the Nobel Prize in Physiology or Medicine. Later, Bordet and Landsteiner were awarded the Nobel Prize in Physiology or Medicine in 1919 and 1930, respectively (Figure 2 and Figure 3).

The early 20th century saw a rivalry between humoral and cellular immunology, with the humoral theory gaining prominence by the early 1900s. The cellular theory of immunity was finally revived in the mid-20th century, particularly through the discovery of the role of lymphocytes and the development of the clonal selection theory, as described in the following chapters [59].

Anaphylaxis and allergy were specified in this period. In 1902, Charles Richet and Paul Portier uncovered anaphylaxis while vaccinating animals against marine toxins (Figure 2) [25]. A second dose caused severe reactions and death, leading them to coin “anaphylaxis”, indicating a lack of defense. Richet was awarded the 1913 Nobel Prize for this finding. In 1903, Nicolas Arthus described the Arthus reaction as a localized immune complex-mediated hypersensitivity response [60]. In 1905, Clemens von Pirquet and Béla Schick first described “serum sickness” [61], noting these symptoms as the body’s response to perceived foreign proteins, like those in horse-derived antitoxins. In the following year, Pirquet coined the term “allergy” (from the Greek words *allos* meaning “other” and *ergon* meaning “work”) to describe an altered reactivity to a substance after a second exposure [26].

### 2.4. Pioneering Immunology of the Mid-20th Century 

In the mid-20th century, there were technological advances that revolutionized immunological research. In the 1920s, Theodor Svedberg invented the ultracentrifuge to separate cellular components (Figure 3) [45]. In 1937, Arne Tiselius developed a practical electrophoresis method for studying biological macromolecules [48], including protein components in blood serum such as albumin and the globulin fractions (α, β, and γ) [62]. Both later won the Nobel Prize in Chemistry, Svedberg in 1926 and Tiselius in 1948, respectively. These techniques were readily utilized to estimate the molecular size of the antibody [63,64], leading to the characterization of an antibody, later named IgM, in 1937 [47]. In 1940, Edwin Cohn successfully purified serum globulins from human plasma, creating a safe injectable material, known as the Cohn process [65,66]. The combination of ultracentrifugation and electrophoresis methods revealed that γ-globulins contained significant amounts of carbohydrates, with different sugar chain compositions found between γ1—and γ2-globulins [67,68,69]. 

Immunofluorescence techniques were first introduced in 1941 by Albert Coons and colleagues, who used fluorescein isothiocyanate (FITC)-labeled antibodies to locate pneumococcal antigens in infected tissues [49]. In 1945, Robin Coombs and colleagues developed a method using anti-human immunoglobulin serum to detect antibodies binding to red blood cells and diagnose hemolytic diseases, known as the Coombs test [51].

In 1960, Rosalyn Yalow and Solomon Berson developed the radioimmunoassay (RIA) method to measure insulin levels in blood [58]. This technique exploited the high specificity of antibodies to measure minute amounts of substances, such as hormones and drugs, in biological fluids. In 1977, Yalow’s work earned her the Nobel Prize in Physiology or Medicine (Figure 4).

While technology provided essential tools, the mid-20th century saw substantial advancement in understanding the humoral and cellular basis of immunity.

In 1923, Michael Heidelberger and Oswald Avery demonstrated that bacterial polysaccharides could trigger an immune response, showing that non-proteins can also act as antigens (Figure 3) [44,70,71]. In 1935, Heidelberger and Forrest Kendall provided conclusive evidence that antibodies are proteins by quantitatively analyzing precipitin reactions [46]. In 1941, Frank Macfarlane Burnet proposed that antibodies could act by blocking receptors on target cells or pathogens, preventing them from interacting with their natural ligands [72].

**Figure 4 antibodies-13-00090-f004:**
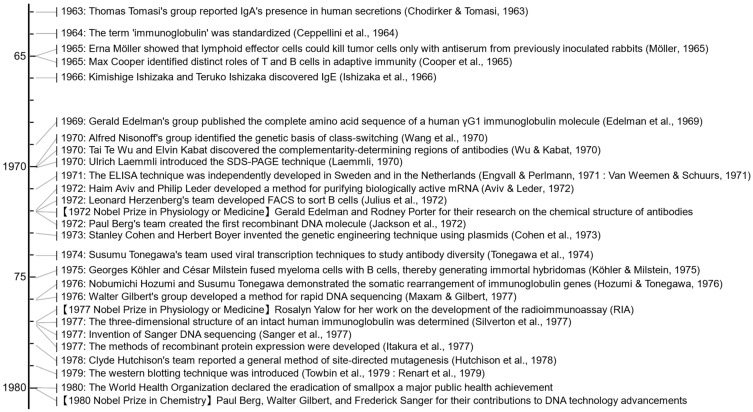
Timeline of selected achievements or events from 1961 to 1980. References in this timeline include works by Chodirker & Tomasi [73], Ceppellini et al. [74], Möller [75], Cooper et al. [8], Ishizaka et al. [76], Edelman et al. [5], Wang et al. [77], Wu & Kabat [78], Laemmli [79], Engvall & Perlmann [80], Van Weemen & Schuurs [81], Aviv & Leder [82], Julius et al. [83], Jackson et al. [84], Cohen et al. [85], Tonegawa et al. [86], Köhler & Milstein [9], Hozumi & Tonegawa [87], Maxam & Gilbert [88], Silverton et al. [89], Sanger et al. [90], Itakura et al. [91], Hutchison et al. [92], Towbin et al. [93], and Renart et al. [94].

In 1943, Mogens Bjørneboe and Harald Gormsen experimentally demonstrated that repeated rabbit immunization resulted in widespread plasma cell proliferation correlated with antibody concentration [50]. Subsequently, in 1947, Astrid Fagraeus discovered the link between antibody production and plasma cells [6], overturning the prevailing theory that the reticuloendothelial system, particularly macrophages, was the primary source of antibodies.

In 1958, Jean Mckenzie identified a thyroid-stimulating factor in sera from thyrotoxic patients that showed a distinct delayed response pattern compared to standard thyrotropin (TSH) [95]. In 1964, Joseph Kriss et al. isolated and characterized this long-acting thyroid stimulator (LATS) as a 7S γ-globulin antibody with unique hormonal properties [96]. LATS was consistently found in patients with pretibial myxedema following treatment for hyperthyroidism, likely representing the initial demonstration of an antibody with hormone-like or agonistic activity.

In 1959, James Gowans demonstrated lymphocyte recirculation by thoracic duct cannulation in rats [97], clarifying how immune responses are coordinated throughout the body and laying the foundation for understanding leukocyte trafficking. He also addressed the importance of T cells in adaptive immunity by showing that the depletion of small lymphocytes severely impairs the ability to mount immune responses [98]. In 1965, Max Cooper and his colleagues identified the thymus and bursa of Fabricius as distinct sites for T and B cell development in chickens (Figure 4) [8], clarifying the separate roles of T cells in cellular immunity and B cells in humoral immunity within the adaptive immune system.

During this period, the existence of Fc receptors of immune cells, along with insights into effector functions such as Antibody-Dependent Cellular Phagocytosis (ADCP) and Antibody-Dependent Cell-Mediated Cytotoxicity (ADCC), began to emerge.

In 1953, Francis Brambell and colleagues found that γ-globulins were selectively transmitted from maternal circulation, relying on the Fc region [99]. He also proposed that a specific receptor, later identified as the neonatal Fc receptor (FcRn), regulates IgG transport in early life and protects it from degradation [100].

In 1963, Boyce Bennett and colleagues demonstrated that isoantibodies or alloantibodies function as specific opsonins, enabling macrophages to recognize and ingest tumor cells [101,102]. In 1966, Arthur Berken and Baruj Benacerraf found that IgG antibodies bind to specific sites on macrophages, enhancing the phagocytosis of opsonized red blood cells independently of complement proteins [103], probably the initial characterization of ADCP [104,105]. In 1965, Erna Möller demonstrated that lymphoid effector cells could kill tumor cells only in the presence of antiserum from previously inoculated rabbits (Figure 4) [75]. It was subsequently discovered that immunoglobulins in the antiserum were essential for this cell-mediated killing [106], which was later termed ADCC [107].

The 1967 study found that human monocytes, macrophages, and certain lymphocytes bind to IgG-coated red blood cells (RBCs) via surface IgG receptors without needing serum complement [108]. Similar membrane receptors were identified on several immune cells, including macrophages [109], monocytes [110], neutrophils [111], basophils [112], and mast cells [113]. Starting around 1972, the binding of immunoglobulins to lymphocytes was characterized through studies on murine B cells [114,115,116,117,118,119] and human B cells [120]. The Fc portion of the immunoglobulin is essential for this binding, leading to the identification of membrane sites as Fc receptors [117].

### 2.5. The Clonal Selection Theory (1950s–1960s) 

The introduction of clonal selection theory in the 1950s brought about another conceptual leap in immunology.

The 1953 paper by Peter Medawar’s group provided evidence for acquired immunological tolerance in mice and established the concept of self/non-self-discrimination (Figure 3) [52]. This work provided a framework for understanding the processes of immune tolerance and autoimmunity. In 1955, Niels Jerne proposed the natural selection theory of antibody formation [53]. This theory suggests that the body naturally produces a diverse array of antibodies, and upon encountering an antigen, those antibodies that bind to it are selectively expanded. He later won the 1984 Nobel Prize in Physiology or Medicine.

David Talmage and Frank Macfarlane Burnet later expanded on these concepts by developing the clonal selection theory [121]. Burnet’s theory, initially documented in 1957 [122] and detailed in 1959 [7], proposed that each lymphocyte is pre-programmed to produce a specific antibody. This theory explained how the immune system could make such a diverse array of antibodies while maintaining self-tolerance.

The theory of clonal selection proposed several key concepts:Each lymphocyte is specific for a single antigen;Upon encountering its specific antigen, a lymphocyte is stimulated to proliferate and differentiate into effector cells;Self-reactive lymphocytes are eliminated during development, ensuring tolerance to self-antigens;Antibody diversity is generated before antigen exposure, not due to antigen exposure.

Several sources have provided experimental support for this theory. In 1956, Theodore Puck, Philip Marcus, and Steven Cieciura developed in vitro techniques for the clonal growth of mammalian cells that provided valuable tools for studying cellular immunity at the single-cell level [54]. These techniques facilitated quantitative cell survival assays and the isolation of mutant cell lines. This work supported the concept of cellular immunity developed by Gustav Nossal and Joshua Lederberg in 1958 [55]. They demonstrated that individual cells produce specific antibodies and that specific antibodies are produced by individual plasma cells rather than by a generalized immune response, outlining the role of antibodies in immune activation.

These findings provide evidence for the clonal nature of the clonal selection theory and antibody responses. Later, Burnet and Medawar achieved the 1960 Nobel Prize in Physiology or Medicine for the discovery of acquired immunological tolerance. The Clonal Selection Theory has profoundly influenced our understanding of the function and dysfunction of the immune system, including autoimmunity [123].

### 2.6. Unraveling Antibody Structure (1950s–1970s)

During the 1950s and 1960s, notable advances were made in the molecular characterization and structural understanding of antibodies.

In the late 1950s, Rodney Porter’s seminal work on the hydrolysis of rabbit γ-globulin with crystalline papain revealed that antibodies comprise three fragments: two identical Fab (fragment antigen-binding) fragments and one Fc (crystallizable) fragment (Figure 1) [124,125]. This discovery demonstrated the modular structure of antibodies, with distinct regions responsible for antigen binding and effector functions. Porter found that each Fab fragment contains an entire light (L) chain and the amino-terminal portion of a heavy (H) chain, confirming that the pairing of an L chain and a H chain forms the antigen-binding site.

Similarly, in 1959, Gerald Edelman published their studies of the structural units of γ-globulin, showing that antibodies are composed of light and heavy polypeptide chains held together by disulfide bonds (Figure 3) [56]. This discovery suggested the basic building blocks of antibodies. In the early 1960s, Edelman’s group continued to study the arrangement of the peptide chains in γ-globulin and showed that antibodies are composed of two identical heavy chains and two identical light chains [126,127]. In 1969, they published the complete amino acid sequence of a human γG1 immunoglobulin molecule arranged in a Y-shaped structure (Figure 4) [5].

These structural studies were supported by the concept of antibody idiotypes, a term initially documented by Jacques Oudin [128] and Michael Kunkel [129] in 1963, and the work of Norbert Hilschmann and Lyman Craig, who in 1965 showed that the N-terminal regions of the light chains were highly variable between different antibodies, while the C-terminal regions were constant [130]. In 1970, Tai Te Wu and Elvin Kabat identified complementarity-determining regions (CDRs) in the variable regions of light and heavy chains critical for antigen binding [78].

The importance of these discoveries was recognized in 1972 when Porter and Edelman were awarded the Nobel Prize in Physiology or Medicine for their separate research on the chemical structure of antibodies. Their proposed model was further refined and verified by the accumulation of crystallographic analyses [131,132], culminating in the three-dimensional structure of an intact human immunoglobulin in 1977 [89].

### 2.7. The Discovery of Antibody Classes and Subclasses (1960s)

The 1960s marked substantial progress in identifying and characterizing different antibody classes and subclasses (Figure 5).

In 1958, Rudolf Gugler, Joseph Heremans, and their colleagues discovered a high-carbohydrate isoform named γA, present in exocrine secretions and migrating in the β-globulin region [133]. In the following year, Jacques Heremans isolated and characterized IgA from human serum (Figure 3) [57]. In 1963, Thomas Tomasi et al. reported its presence in human secretions (Figure 4) [73]. In 1964, the term ‘immunoglobulin’ was standardized in an article published in the Bulletin of the World Health Organization [74]. The same year, David Rowe and John Fahey identified IgD [134,135]. Their work was published on the following New Year’s Day.

A breakthrough came in 1966 when Kimishige Ishizaka and Teruko Ishizaka identified IgE as a carrier of receptor activity, which is critical to understanding allergic reactions [76,136]. Ishizaka’s team demonstrated that IgE was responsible for the immediate hypersensitivity reactions previously attributed to “reagins” [137,138] and confirmed it as a new immunoglobulin identical to the myeloma protein reported by Gunnar Johansson and Hans Bennich in 1967 [139,140].

The discovery of antibody subclasses complemented the identification of new antibody classes. In 1964, three studies demonstrated that human 7S γ-globulin (IgG) was not homogeneous but consisted of multiple distinct subclasses [141,142,143]. In 1965, William Terry and Fahley described the four subclasses of human IgG (IgG1, IgG2, IgG3, and IgG4) [144]. That same year, William Dreyer and Claude Bennett proposed the important concept of separate genes for the variable (V) and constant (C) regions of antibody chains, providing the groundwork for understanding antibody diversity [145].

Other advances in the field included the identification of the mechanisms by which B cells switch isotype, a process known as class-switch recombination (CSR). In 1970, Alfred Nisonoff and colleagues provided the first evidence of isotype switching [77], showing that monoclonal immunoglobulins G and M from a single patient shared idiotypic determinants and had identical amino acid sequences. The presence of a shared sequence in two antibodies of different isotypes suggests a common origin, indicating a genetic switching event during the expansion of that specific B cell clone. Subsequent research elucidated that during B-cell differentiation, the joining of V_L_ (light chain variable) and J_L_ (light chain joining) regions, and C_H_ (heavy-chain constant regions) switching, are crucial for producing diverse antibodies, as demonstrated by changes in gene organization in mature B cells [146]. T cell cytokines play an important role in influencing CSR, with IFN-γ promoting IgG2a production and IL-4 driving the synthesis of IgG1 and IgE, thereby generating functional diversity of antibodies and contributing to efficient immune responses [147].

In 2000, Tasuku Honjo and colleagues identified activation-induced cytidine deaminase (AID) as a critical enzyme in class switching from IgM to other isotypes such as IgA, IgG, and IgE (Figure 6) [148]. Together, these discoveries increased our understanding of the diversity and specificity of the humoral immune response.

### 2.8. The Genetic Basis of Antibody Diversity (1970s) 

In the 1970s, molecular biology and immunology advanced remarkably as new biotechnologies blossomed.

In 1970, Ulrich Laemmli introduced the sodium dodecyl sulfate-polyacrylamide gel electrophoresis (SDS-PAGE) technique (Figure 4) [79]. In 1971, the enzyme-linked immunosorbent assay (ELISA) technique was independently developed by Peter Perlmann and Eva Engvall in Sweden [80] and Anton Schuurs and Bauke van Weemen in the Netherlands [81] to detect specific antigens. In 1972, Haim Aviv and Philip Leder developed a method for purifying biologically active messenger RNA (mRNA) [82]. In the same year, Leonard Herzenberg and his team developed fluorescence-activated cell sorting (FACS) and used it to isolate antigen-binding B cells [83,162], with the initial concept reported in 1965 [163]. Paul Berg and colleagues created the first recombinant DNA molecule by combining DNA from different species [84]. Around 1973, Stanley Cohen and Herbert Boyer invented the genetic engineering technique of creating recombinant DNA molecules and introducing them into bacteria using plasmids [85]. Around 1976, Allan Maxam and Walter Gilbert developed a method for rapid DNA sequencing [88]. In 1977, Sanger DNA sequencing [90] and the methods of recombinant protein expression [91] were introduced. Berg, Gilbert, and Sanger won the 1980 Nobel Prize in Chemistry. In 1978, Clyde Hutchison and colleagues reported a general method of site-directed mutagenesis [92], leading to Hutchison receiving the 1993 Nobel Prize in Chemistry (Figure 6). In 1979, the Western blotting technique was introduced (Figure 4) [93,94].

During this time, scientists were still struggling to explain the diversity of antibodies. The debate centered on whether different antibodies were encoded by distinct genes (germline theory) or resulted from variation during replication (somatic differentiation theory). In 1967, Gerald Edelman and Joseph Gally proposed a solution, suggesting that a few duplicated genes in tandem arrays with point mutations undergo somatic crossover during lymphocyte development [164]. This process, facilitated by homologous regions and gene proximity, creates new sequences with different mutations. However, their work, based on observations of hot spots in the variable region of Bence Jones proteins, did not fully explain the lack of hypervariable segments in the constant region.

In 1974, Susumu Tonegawa, who later won the 1987 Nobel Prize in Physiology or Medicine, and his colleagues used viral transcription techniques to study antibody diversity (Figure 4) [86]. Their DNA hybridization experiments with different κ chain mRNAs showed that germline genes alone could not account for the observed antibody diversity. In 1976, Nobumichi Hozumi and Tonegawa provided compelling evidence for the somatic rearrangement of immunoglobulin genes [87]. They showed that the V (variable) and C (constant) genes are segregated in mouse embryonic cells but join to form a continuous V-C gene in differentiated lymphocytes, a finding later confirmed by R-loop mapping in 1978 [165]. In 1983, Tonegawa detailed the mechanisms responsible for antibody diversity, highlighting somatic recombination and mutation [10].

The discovery of the joining (J) and diversity (D) regions in the immunoglobulin heavy (H) chain furthered our understanding of antibody diversity. In 1981, Frederick Alt and colleagues elucidated the rearrangement process in cells transformed by the Abelson murine leukemia virus (Figure 6) [149]. They observed several distinct rearrangements near the J_H_ regions but not the J_L_ (light) regions, suggesting that heavy chain rearrangement precedes light chain rearrangement. Alt indicated that the addition of light chains to a fully assembled heavy chain might help assess the success of the light chain gene rearrangement.

The focus then shifted to identifying the recombination machinery, including V(D)J recombination. In 1989, David Schatz and Marjorie Oettinger, working in David Baltimore’s laboratory, isolated recombination-activating gene 1 (RAG-1) [156]. In 1990, they demonstrated that RAG-1, together with RAG-2, initiates V(D)J recombination by making DNA double-strand breaks, confirming that RAG-1 is a critical component of the recombinase complex [166]. Later, in 1995, it was reported that RAG-1 works with RAG-2 to make the DNA double-strand breaks that initiate V(D)J recombination [167].

It is now known that, beyond V(D)J recombination, antibody diversity is generated through CSR and somatic hypermutation (SHM), both mediated by AID [168]. A growing body of evidence highlights the importance of additional players, including transcription factors by controlling chromatin accessibility, DNA repair enzymes by generating DNA breaks for recombination, and epigenetic modifications by regulating chromatin structure [169].

### 2.9. Monoclonal Antibodies: From Bench to Bedside (1970s)

The development of monoclonal antibody technology in the 1970s marked a turning point in immunology and biotechnology.

In 1973, Dick Cotton, working in César Milstein’s laboratory, demonstrated that fusing two myeloma cell lines could produce a hybrid cell line capable of secreting antibodies from both parent lines [170]. Milstein’s collaboration with Georges Köhler, who joined his laboratory in 1974, led to a breakthrough. Despite existing methods for cloning single antibody-producing B cells, such as those developed by Norman Klinman, these methods had limitations such as low yields and short cell lifetimes [171]. Köhler and Milstein’s solution was to create a hybridoma by fusing myeloma cells with spleen cells from an immunized mouse (Figure 4) [9]. This hybridoma combined the immortality of myeloma cells with the antibody specificity of spleen cells and produced large quantities of monoclonal antibodies. The efficient production of hybridomas became possible [172], and these techniques gradually led to widespread interest in and the commercialization of monoclonal antibody technology. Milstein and Köhler were awarded the 1984 Nobel Prize in Physiology or Medicine for their work (Figure 6).

In the early 1980s, monoclonal antibodies transitioned from research tools to potential therapeutic agents. The first administration of monoclonal antibodies to patients with lymphocytic malignancies was reported in 1980 [173] but did not produce substantial clinical results. The breakthrough came in 1982 when Ronald Levy and colleagues successfully treated a patient with chemotherapy-resistant B-cell lymphoma with anti-idiotype monoclonal antibodies [150,174]. The patient remained in remission for more than 30 years after this trial. In 1986, muromonab-CD3 (OKT3) [175] became the first monoclonal antibody approved by the FDA to prevent kidney transplant rejection. However, rodent-derived monoclonal antibodies such as OKT3 were immunogenic in humans, causing excessive immunosuppression [176], and were voluntarily withdrawn from the U.S. market in 2010 due to safer alternatives and reduced demand [176,177,178].

### 2.10. Antibody Humanization and Clinical Advances (1980s)

To address antibody immunogenicity, in 1984, Vernon T. Oi and colleagues [151] and Marc Shulman and colleagues [152] developed “chimeric antibodies” of IgG and IgM, respectively (Figure 6). They achieved this by combining mouse variable regions with human constant domains to minimize immune responses while maintaining the ability to bind to antigens. In 1986, Gregory Winter’s team developed “humanized antibodies”, a technique that involves grafting complementarity-determining regions (CDRs) from mouse antibodies onto human frameworks [11]. This process significantly reduces human anti-mouse antibody (HAMA) responses and improves therapeutic suitability. In addition, the development of phage display technology by George Smith in 1985 [153], the production of Fab fragments through bacterial expression systems [154], the invention of a single-chain variable fragment (scFv) linking the V_L_ and V_H_ regions with an optimal linker in 1988 [179,180], sequence randomization first reported in 1990 [157], and the development of scFv display techniques [158] and methods for selecting human antibodies from phage display repertoires [181] have facilitated the rapid identification and optimization of antibodies with high specificity and affinity. Later, George Smith and Gregory Winter received the 2018 Nobel Prize in Chemistry. 

These advances paved the way for a new generation of therapeutic antibodies. For example, Herman Waldmann’s team developed CAMPATH-1 [182], which targets the mature lymphocyte antigen CD52 [183]. CAMPATH-1H, the humanized version, was first tested in humans in 1988 [155], eventually leading to the FDA approval of lemtrada (alemtuzumab) for multiple sclerosis in 2014 [184].

In 1997, the FDA approved Zenapax (daclizumab), the first humanized monoclonal antibody, for the prevention of acute organ rejection in kidney transplant patients [185]. That same year, Rituxan (rituximab), which targets the B-cell lineage marker CD20, was approved for treating B-cell non-Hodgkin’s lymphoma [186,187], becoming the first chimeric monoclonal antibody for cancer treatment. Rituximab mediates ADCC and CDC and induces apoptosis in B-cell lymphomas through direct signaling, providing an initial demonstration of direct signaling activity in antibodies [186,187]. In 1998, Herceptin (trastuzumab), which targets HER2, was approved for HER2-positive breast cancer [188,189].

In 2002, the FDA approved the first radioimmunotherapy monoclonal antibody, Zevalin (^90^Yttrium ibritumomab tiuxetan), for the treatment of certain types of B-cell non-Hodgkin lymphoma (NHL) (Figure 7). The initial clinical trial that led to this approval was reported in 1993 [190]. To produce fully human antibodies, various biotechnological methods are used, such as genetically engineered phage display [191] and transgenic mouse models [192,193]. In 2002, the FDA approved Humira (adalimumab), the first human antibody derived from phage display, for the treatment of rheumatoid arthritis by targeting TNF-α. Similarly, in 2006, Vectibix (panitumumab), the first fully human antibody generated from transgenic mice, was approved for the treatment of metastatic colorectal cancer by targeting the Epidermal Growth Factor Receptor (EGFR).

Targeting angiogenesis is effective in several tumor types [194]. In 2004, Avastin (bevacizumab) was approved for metastatic colorectal cancer [195]. Bevacizumab inhibits vascular endothelial growth factor (VEGF), a concept first described by Napoleone Ferrara and colleagues in 1993 [196]. Bevacizumab is currently used in combination regimens to treat several cancers, including colorectal cancer, non-small cell lung cancer, and glioblastoma [195].

In the 2000s, modifications to the Fc region—including hinge regions and the glycosylation of antibodies—known as Fc engineering, were envisaged based on structural and biological studies to modulate effector functions, increase antibody half-life, and improve pharmacokinetics [197,198,199]. In 2003, the lack of fucose in IgG1 antibodies, namely afucosylated antibodies, was reported to enhance ADCC, showing over 50-fold greater activity than fucosylated antibodies [200]. POTELIGEO (mogamulizumab-kpkc), approved in 2012, is the first approved glycol-engineered, afucosylated antibody [201]. It is a humanized monoclonal antibody that targets CCR4 for the treatment of adult patients with relapsed or refractory mycosis fungoides (MF) or Sézary syndrome (SS). Engineered antibody variants with improved binding affinity to FcRn, such as YTE (M252Y/S254T/T256E) [202] and LS (M428L/N434S) [203] mutations, have been developed to extend serum half-life, allowing for less frequent dosing. In 2018, Ultomiris (ravulizumab-cwvz) was approved for paroxysmal nocturnal hemoglobinuria (PNH), containing the LS mutations and offering a four-fold longer dosing interval than its predecessor, Soliris (eculizumab) [204].

Now, the field of immunotherapy has advanced with the introduction of antibody-drug conjugates (ADCs) [205,206], bispecific antibodies (bsAbs) [207,208] such as T cell engagers (TCEs) like T-cell bispecific antibodies (TCBs or BiTEs) [209,210], and chimeric antigen receptor T-cell therapy (CAR-T) [211,212].

ADCs, which combine an antibody specific to cancer cells with a cytotoxic drug [205], allow targeted delivery of the drug, minimizing collateral damage to healthy cells. In 2000, the FDA approved Mylotarg (gemtuzumab ozogamicin), the first ADC to treat relapsed acute myeloid leukemia (AML) [213].

bsAbs are engineered to bind to two different antigens simultaneously. For example, TCBs are designed to bind both a tumor antigen and the T cell receptor. Unlike other immune cells, T cells do not express Fc receptors and are not activated by conventional IgG antibodies through effector functions. The dual binding of TCBs facilitates the proximity of T cells to cancer cells, aiding in targeted cell killing [208]. The first approved bsAb was Removab (catumaxomab), a trifunctional rat-mouse chimeric antibody designed to bind to epithelial cellular adhesion molecule (EpCAM) and CD3ε [214]. The double epitope specificity drives the interaction between cancer cells expressing EpCAM and activated T cells, while the Fc-region binding of catumaxomab with Fcγ receptors enables immune accessory cells to kill the target. In 2009, the European Medicines Agency (EMA) approved this bsAb for the treatment of malignant ascites (Figure 7). Despite its therapeutic potential, catumaxomab was voluntarily withdrawn from the U.S. market in 2013 and the EU market in 2017 for commercial reasons. In 2014, the FDA approved Blincyto (blinatumomab), the second bsAb, to treat relapsed or refractory B-cell precursor acute lymphoblastic leukemia [215,216].

In CAR-T cell therapy, a patient’s T cells are genetically engineered to express chimeric antigen receptors (CARs) targeting cancer cells. The structure of a typical CAR includes an extracellular scFv for antigen recognition and intracellular domains like CD3ζ for T cell activation [211]. These components work together to enable CAR-T cells to target and destroy cancer cells specifically. In 2017, the FDA approved the first CAR-T cell therapy, Kymriah (tisagenlecleucel), for certain types of blood cancers [217].

These innovative therapies, each with different mechanisms and potential benefits, are advancing cancer care by providing effective options for patients with limited therapeutic alternatives.

### 2.11. Immune Checkpoint Inhibitors (1990s)

In the 1990s, a new paradigm in cancer treatment emerged: immune checkpoint inhibition. This approach was based on an evolving understanding of how the immune system is regulated and how cancer cells use these mechanisms to evade immune detection. In 1992, Tasuku Honjo and colleagues discovered PD-1 (programmed cell death protein 1), a critical immune checkpoint molecule (Figure 6) [159]. PD-1′s function was initially unclear, but subsequent research revealed its role in regulating T-cell responses [218]. In 1999, PD-1 knockout mice were shown to develop autoimmune diseases such as lupus-like proliferative arthritis and glomerulonephritis [219], suggesting its role as a negative regulator of immune responses.

At the same time, James Allison was studying another checkpoint molecule, CTLA-4 (cytotoxic T-lymphocyte-associated protein 4). Allison’s team demonstrated that CTLA-4 acts as a “brake” on T cells and hypothesized that blocking CTLA-4 could unleash the immune system to attack cancer cells [220]. In 1996, Allison and colleagues showed that antibodies blocking CTLA-4 could induce tumor rejection in mice [161], suggesting the potential to treat cancer by modulating the immune system rather than directly targeting cancer cells.

These discoveries led to the development of checkpoint inhibitor therapies [221]. In 2011, a major milestone was achieved with the FDA’s approval of Yervoy (ipilimumab) (Figure 7) [222,223], an anti-CTLA-4 antibody, for treating melanoma. The success of ipilimumab was followed by the development and approval of anti-PD-1 antibodies. In 2014, the FDA approved Keytruda (pembrolizumab) and Opdivo (nivolumab) for the treatment of melanoma [224,225]. These immune checkpoint inhibitors (ICIs) have shown impressive results, with some patients experiencing long-lasting remissions. The impact of ICIs extended beyond melanoma. In the following years, ICIs were approved for a growing list of cancers, including lung, kidney, and bladder [226]. ICIs also showed promise in combination therapies with other immunotherapies and traditional cancer treatments, such as chemotherapy and radiation [227]. In 2018, James Allison and Tasuku Honjo were awarded the Nobel Prize in Physiology or Medicine for their pioneering work in cancer immunotherapy.

ICIs have revolutionized cancer treatment, improving patient outcomes and survival rates. In colorectal cancer (CRC), ICIs have shown promise for patients with microsatellite instability-high (MSI-H) or deficient mismatch repair (dMMR) advanced CRC [228]. The expanding role of ICIs in oncology is further exemplified by recent FDA approvals for biliary tract cancer (BTC). In 2022, Imfinzi (durvalumab) received approval, followed by pembrolizumab in 2023, both in combination with gemcitabine and cisplatin for BTC treatment [229].

### 2.12. Cytokines Inhibitors (1990s)

Monoclonal antibodies targeting cytokines have transformed autoimmune disease care. The hypothesis that cytokines play a critical role in autoimmune diseases emerged in the early 1980s from immunohistological data showing an upregulated expression of major histocompatibility complex class II in autoimmune tissues [230,231,232]. Marc Feldmann speculated that cytokines might control this effect. In collaboration with Ravinder Maini and Fionula Brennan, in 1989, they demonstrated the presence of several pro-inflammatory cytokines in inflamed joints [233]. They showed that blocking TNF in cultures of human rheumatoid tissue stopped the production of interleukin-1 (IL-1β) and other cytokines, suggesting that TNF is a “master regulator” in rheumatoid arthritis. In 1993, Feldmann and Maini conducted a small clinical trial to test a chimeric TNF-specific monoclonal antibody, which was later named Remicade (infliximab) (Figure 6) [160]. Nearly all patients reported rapid improvements in pain, fatigue, and mobility, as well as reduced inflammation. Infliximab was approved for Crohn’s disease in 1998. Although the effects were short-lived, subsequent retreatments showed consistent responses [234]. Long-term studies have confirmed the efficacy and durability, especially in combination with methotrexate [235].

A handful of TNF-specific monoclonal antibodies are currently approved for treating rheumatoid arthritis and other diseases such as juvenile idiopathic arthritis, psoriatic arthritis, ankylosing spondylitis, ulcerative colitis, Crohn’s disease, and psoriasis [236]. One of the most widely used medications is adalimumab. TNF plays a vital role in these diseases by recruiting leukocytes and upregulating adhesion molecules and chemokines. However, not all patients benefit from TNF inhibitors [237]. For example, some patients with juvenile idiopathic arthritis respond better to Actemra (tocilizumab) (Figure 7) [238], an IL-6 receptor antibody developed by Tadamitsu Kishimoto.

Recently, IL-17A-specific antibodies were developed for the treatment of psoriasis. IL-17A, produced by T cells and initially discovered in 1993 [239], plays a crucial role in psoriasis by stimulating keratinocytes to secrete chemokines and other mediators [240,241]. Cosentyx (secukinumab), the first IL-17A-specific antibody approved in 2015 (Figure 7), has shown significant efficacy [242]. Other IL-17A blockers, such as Taltz (ixekizumab) in 2016 and Siliq (brodalumab) in 2017, have been approved [243]. In addition, Bimzelx (bimekizumab-bkzx), which targets both IL-17A and IL-17F within the same variable region—distinguishing it from bispecific antibodies—has shown promising results in clinical trials [244,245] and was approved in 2023.

The success of TNF blockade has led to the development of multiple cytokine-targeting antibody drugs. New biologics targeting IL-23, such as Tremfya (guselkumab) and Skyrizi (risankizumab-rzaa), have been approved, providing more options for effectively treating psoriasis [246]. CAR T-cell therapy has also shown promise in treating systemic autoimmune diseases, such as systemic lupus erythematosus and systemic sclerosis, by targeting and depleting autoreactive B cells, leading to remarkable improvements or remissions [247].

## 3. Future Directions

The field of immunology and antibody research has evolved significantly since the early days of smallpox vaccination. Antibody research provides critical solutions to global health challenges, including emerging infectious diseases and antibiotic-resistant bacteria. In 2020, the COVID-19 pandemic accelerated the development and licensure of mRNA vaccines that stimulate the production of antibodies against the SARS-CoV-2 spike protein [248]. This rapid development and success represent a new milestone in immunology and vaccinology. The 2023 Nobel Prize in Physiology or Medicine was awarded to Katalin Karikó and Drew Weissman for their discoveries concerning nucleoside base modifications, which enabled the development of effective mRNA vaccines (Figure 7) [249,250,251].

Several promising developments mark the future of antibody research:Antibody therapies are being investigated for a broader range of diseases, including neurodegenerative and metabolic disorders [252,253,254]. In addition, they are being considered as alternatives to traditional vaccines, offering new options for disease prevention [255].Researchers are exploring pairing antibodies with other treatment modalities, like small-molecule drugs or cell therapies, to create more effective treatments for complex diseases [205,256]. For example, combining gemcitabine and cisplatin with a monoclonal antibody demonstrates this approach.Advances in delivery methods, such as oral and transdermal delivery, are being explored to make antibody therapies more accessible and convenient for patients [257,258].Innovations such as nanobodies, cattle-derived knob domains, intrabodies, and multispecific antibodies are expanding the therapeutic potential of antibody-based drugs [259,260,261]. These formats offer unique properties, including improved tissue penetration and the ability to target multiple antigens simultaneously.Advances in genomics, proteomics, and metabolomics are enabling the development of personalized antibody therapies customized to individual patients’ genetic and acquired profiles, including specific disease characteristics and body microbiome, to enhance treatment effectiveness and minimize side effects [262,263].The integration of AI and machine learning into antibody design accelerates the discovery and optimization of new therapeutics [264]. These technologies facilitate the prediction of antibody properties and the design of new antibodies from scratch [265]. In 2024, the Nobel Prize in Chemistry was awarded to David Baker for computational protein design, and to Demis Hassabis and John Jumper for protein structure prediction.

With each development and technological advance, we come closer to unlocking the immune system’s full potential in treating and preventing diseases.

## Figures and Tables

**Figure 1 antibodies-13-00090-f001:**
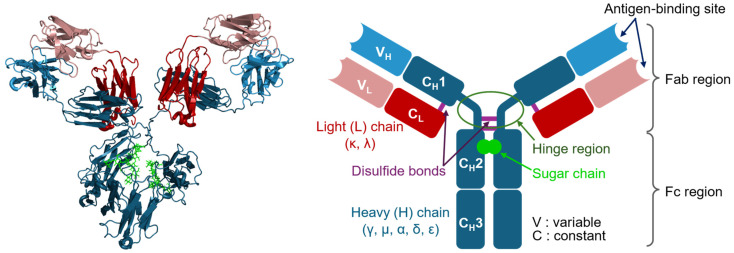
Crystal structure and schematic diagram of immunoglobulin G. The crystal structure of immunoglobulin G1 (PDB code: 1IGY) is illustrated using a ribbon model on the left and a schematic diagram on the right [12]. The light variable (V_L_), light constant (C_L_), heavy variable (V_H_), and heavy constant (C_H_) regions are depicted in light red, dark red, light blue, and dark blue, respectively. Carbohydrate residues attached to the Fc region are shown in green using stick models.

**Figure 2 antibodies-13-00090-f002:**
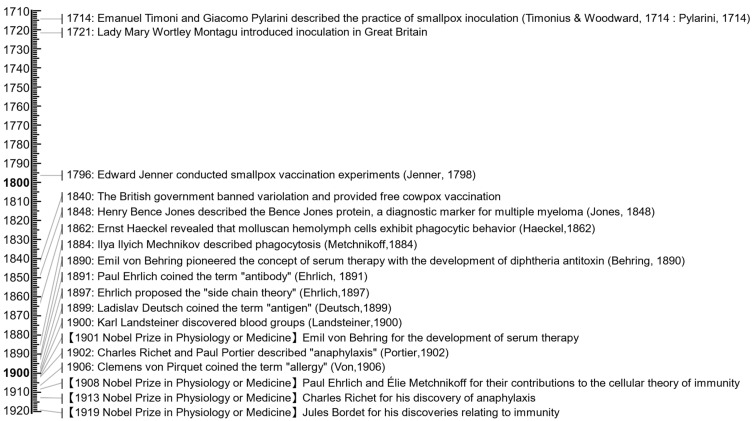
Timeline of selected achievements or events from the 18th century to the early 20th century. References in this timeline include works by Timonius & Woodward [14], Pylarini [15], Jenner [16], Jones [17], Haeckel [18], Metchnikoff [19], Behring [20], Ehrlich [21,22], Deutsch [23], Landsteiner [24], Portier [25], and Von [26].

**Figure 3 antibodies-13-00090-f003:**
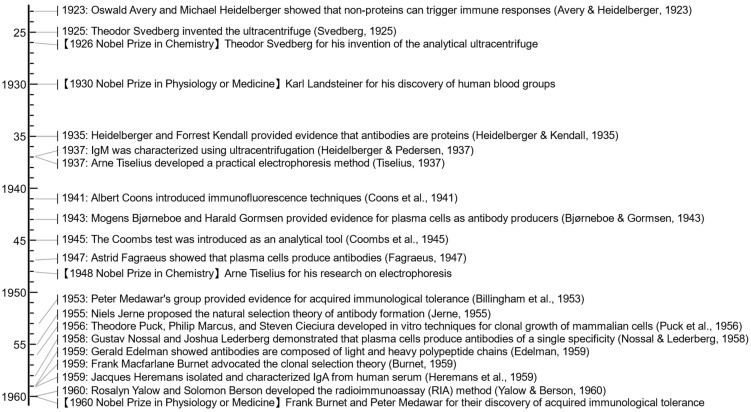
Timeline of selected achievements or events from 1921 to 1960. References in this timeline include works by Avery & Heidelberger [44], Svedberg [45], Heidelberger & Kendall [46], Heidelberger & Pedersen [47], Tiselius [48], Coons et al. [49], Bjørneboe & Gormsen [50], Coombs et al. [51], Fagraeus [6], Billingham et al. [52], Jerne [53], Puck et al. [54], Nossal & Lederberg [55], Edelman [56], Burnet [7], Heremans et al. [57], and Yalow & Berson [58].

**Figure 5 antibodies-13-00090-f005:**
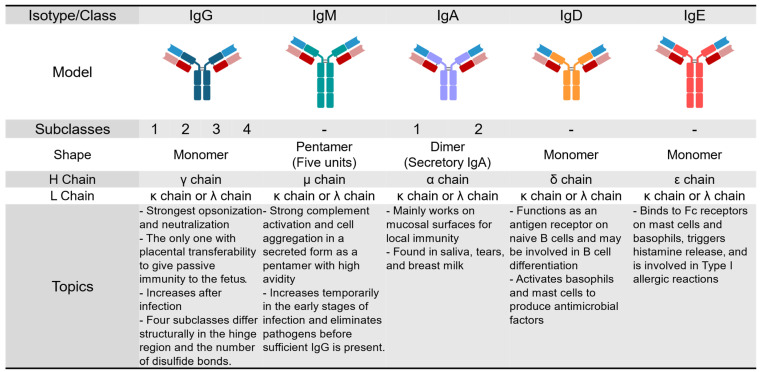
Comparison of immunoglobulin isotypes: structure and function. This diagram shows the five main classes of immunoglobulins, each with a specific role in the immune system. All immunoglobulins have a basic structure that is made up of two heavy chains and two light chains arranged in a Y shape. The type of heavy chain is called the isotype, and it determines the antibody’s class and its specific function. The Fc region of the antibody molecule, found at the base of the Y shape, is essential for activities such as opsonization (labeling pathogens for destruction), complement activation (a sequence of proteins that helps eliminate pathogens), and binding to various cell receptors. The variable region at the tips of the Y shape is responsible for recognizing and binding to specific antigens or foreign substances. The five classes of immunoglobulins—IgG, IgM, IgA, IgD, and IgE—each have distinct characteristics in terms of structure, function, and expression, as briefly described in the topics section.

**Figure 6 antibodies-13-00090-f006:**
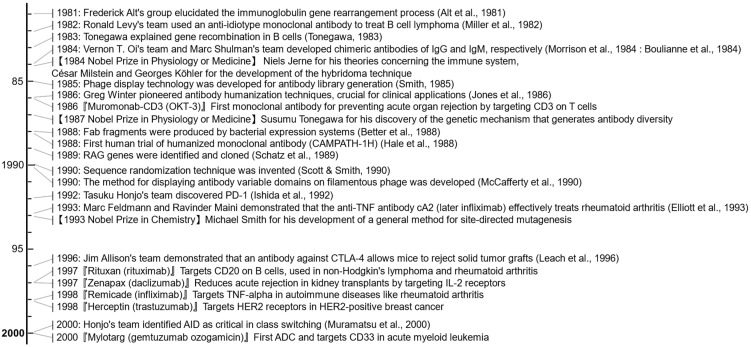
Timeline of selected achievements or events from 1981 to 2000. References in this timeline include works by Alt et al. [149], Miller et al. [150], Tonegawa [10], Morrison et al. [151], Boulianne et al. [152], Smith [153], Jones et al. [11], Better et al. [154], Hale et al. [155], Schatz et al. [156], Scott & Smith [157], McCafferty et al. [158], Ishida et al. [159], Elliott et al. [160], Leach et al. [161], and Muramatsu et al. [148].

**Figure 7 antibodies-13-00090-f007:**
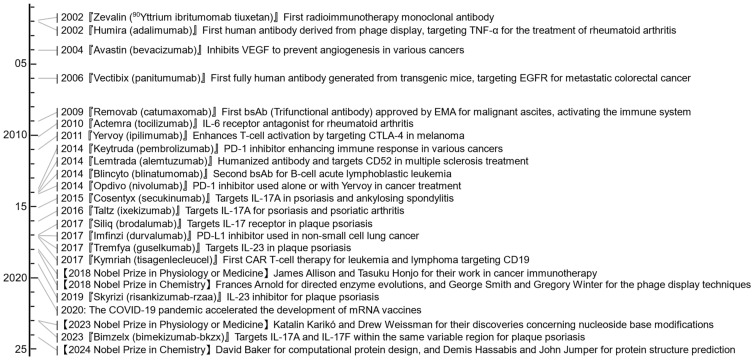
Timeline of selected achievements or events from 2001.

## Data Availability

Not applicable.

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
