# Peer review of "A Brief Chronicle of Antibody Research and Technological Advances"

_2073-4468, 2024, doi:10.3390/antib13040090_

Round 1

Reviewer 1 Report

Comments and Suggestions for Authors

In their review article “A Brief Chronicle of Antibody Research and Technological Advances” Kazutaka Araki and Ryota Maeda summarize historical key findings and the development of antibody research towards modern immunotherapy. The manuscript is well written and gives a comprehensive overview. However, some major points could be addressed to further improve the manuscript:

1.       In the legend of Fig. 1 the antibody subclass should be specified regarding the existing structural differences in the hinge regions and disulfide bonds. It should be considered that the ribbon model shows an IgG2 antibody, which typically has more than two disulfide bonds between the heavy and the light chain, in contrast to IgG1, which typically has two bonds. It would be more consistent with the schematic drawing to depict a ribbon drawing for IgG1. For the ribbon drawing the cited manuscript should be included in the reference list (in addition to citing the pdb file in the figure legend).

2.       One key aspect of therapeutic antibodies is that they can mediate various effector functions such as ADCC, ADCP, CDC, receptor blockade or direct signaling. The paper would benefit if those were briefly introduced.

3.       L. 408: It would be important to introduce the term “humanized antibody” to clarify the difference between humanized and chimeric antibodies.

4.       It would be important to explain technologies that allow the generation of “fully human antibodies” and to give examples (e.g. adalimumab).

5.       L. 446. It would be important for the reader to understand that in this application bispecific antibodies facilitate to trigger T cells, which normally do not express Fc receptors and are not activated by canonical IgG antibodies.

6.       L. 453. It may be worth mentioning that catumaxomab has been withdrawn.

7.       L. 456. The structure of a typical CAR should be explained in more detail, indicating the use of an scFv for antigen binding and the coupled intracellular domains for T cell activation.

8.       Besides bsAbs, ADC and CART, Fc engineering is another important approach, and there are several examples for Fc engineered antibodies that have been approved for clinical application. Therefore, it would be important to briefly introduce Fc engineering as a strategy to improve antibodies by increasing or reducing Fc effector functions, hinge stabilization or half-life extension.

Author Response

We are grateful for your constructive feedback and the opportunity to improve our manuscript. We hope that the revisions adequately address the concerns raised and strengthen the overall quality of our manuscript.

Comments 1: In the legend of Fig. 1 the antibody subclass should be specified regarding the existing structural differences in the hinge regions and disulfide bonds. It should be considered that the ribbon model shows an IgG2 antibody, which typically has more than two disulfide bonds between the heavy and the light chain, in contrast to IgG1, which typically has two bonds. It would be more consistent with the schematic drawing to depict a ribbon drawing for IgG1. For the ribbon drawing the cited manuscript should be included in the reference list (in addition to citing the pdb file in the figure legend).

 Response 1: Thank you for the imperative information. Figure 1 and its figure legend were revised and updated as advised. Since the four IgG subclasses have distinct antibody structures, the following comment was added to Figure 6 in the IgG part: “Four subclasses differ structurally in the hinge region and the number of disulfide bonds”.

Comments 2: One key aspect of therapeutic antibodies is that they can mediate various effector functions such as ADCC, ADCP, CDC, receptor blockade or direct signaling. The paper would benefit if those were briefly introduced.

 Response 2: We totally agree that effector functions such as ADCC, ADCP, and CDC are essential for the therapeutic efficacy of antibodies in various clinical applications. While receptor blockade, agonist activity, and direct signaling are not typically classified as traditional effector functions, they represent significant mechanisms through which antibodies can achieve therapeutic effects. The revisions introducing the importance of these mechanisms have been integrated throughout the text as follows: CDC (Lines: 107-111, red text), ADCP (Lines: 204-216, red text), receptor blockage (Lines: 179-181, red text), and direct signaling (Lines: 470-472, red text).

Comments 3: L. 408: It would be important to introduce the term “humanized antibody” to clarify the difference between humanized and chimeric antibodies.

 Response 3: Thank you for your critical advice regarding the importance of clarifying the difference between "chimeric antibodies" and "humanized antibodies". We agree that this distinction is crucial to understand the evolution of antibody engineering. We have made the revisions to address this comment (Lines: 448-451, red text).

Comments 4: It would be important to explain technologies that allow the generation of “fully human antibodies” and to give examples (e.g. adalimumab).

Response 4: Thank you for this valuable suggestion. We have expanded the main text to include the technologies for generating fully human antibodies (Lines: 478-484, red text).

Comments 5: L. 446. It would be important for the reader to understand that in this application bispecific antibodies facilitate to trigger T cells, which normally do not express Fc receptors and are not activated by canonical IgG antibodies.

 Response 5: As suggested, we revise the text to introduce that “bispecific antibodies facilitate to trigger T cells, which normally do not express Fc receptors and are not activated by canonical IgG antibodies” (Lines: 516-519, red text)

Comments 6: L. 453. It may be worth mentioning that catumaxomab has been withdrawn.

 Response 6: Thanks for the additional information. We revised the text to add the information of the withdrawal (Lines: 525-527, red text).

Comments 7: L. 456. The structure of a typical CAR should be explained in more detail, indicating the use of an scFv for antigen binding and the coupled intracellular domains for T cell activation.

 Response 7: Thanks for the valuable information. We revised the text to add the information of the structure of a typical CAR (Lines: 531-534, red text).

Comments 8: Besides bsAbs, ADC and CART, Fc engineering is another important approach, and there are several examples for Fc engineered antibodies that have been approved for clinical application. Therefore, it would be important to briefly introduce Fc engineering as a strategy to improve antibodies by increasing or reducing Fc effector functions, hinge stabilization or half-life extension.

 Response 8: Thank you for this valuable suggestion. We have expanded our text to include Fc engineering as an important strategy for antibody optimization (Lines: 491-505, red text). Also, we have shortly included the historic discovery of Fc receptors (Lines: 220-227, red text).

Reviewer 2 Report

Comments and Suggestions for Authors

Congratulations on a well researched and comprehensive review of antibody research from ancient China through the 1700s to present day.

Minor points for consideration:

412, reference given is for expression of Fv in E.coli.  Consider using Better et al., 1988 Science 240, 1041 for Fab expression.

444, reference to another review should be changed for an original paper describing Mylotarg, e.g. Hamann et al., 2001 in Bioconjugate Chemistry

530, and Figure 7 402, could highlight Bimzelx is dual specific for IL-17A and F within the same variable region, to differentiate it from bispecific antibodies. 

562, Bovine knob-derived antibody fragments should be acknowledged as the world's smallest antibody fragments, which are uniquely suited to the applications mentioned, e.g. Kuravsky et al., 2024 Frontiers in Immunology.  

Author Response

We are grateful for your constructive feedback and the opportunity to improve our work. We hope that the revisions adequately address the concerns raised and strengthen the overall quality of our manuscript.

Comments 1: 412, reference given is for expression of Fv in E.coli.  Consider using Better et al., 1988 Science 240, 1041 for Fab expression.

Response 1: Thank you for your valuable suggestion regarding the reference for Fab expression. The reference [172] was revised as suggested.

Comments 2: 444, reference to another review should be changed for an original paper describing Mylotarg, e.g. Hamann et al., 2001 in Bioconjugate Chemistry

Response 2: As advised, we cited the original article [210] instead of citing the review article.

Comments 3: 530, and Figure 7 402, could highlight Bimzelx is dual specific for IL-17A and F within the same variable region, to differentiate it from bispecific antibodies. 

Response 3: As advised, we revised the text (Lines: 607-608, red text) and Figure 7 (Line: 442).

Comments 4: 562, Bovine knob-derived antibody fragments should be acknowledged as the world's smallest antibody fragments, which are uniquely suited to the applications mentioned, e.g. Kuravsky et al., 2024 Frontiers in Immunology.  

Response 4: Thank you for your valuable comment regarding bovine knob-derived antibody fragments. We have added the following text to line 640: “cattle-derived knob domains”. We have also included the suggested reference [261].

Reviewer 3 Report

Comments and Suggestions for Authors

This review, humbly titled “a brief chronicle,” is an exhaustive, extensively researched, and informative recounting of the history of antibody research, from the very roots of immunology in the 18th century up to the newest advances of the 2020s. I enjoyed reading it and recommend it for publication as is.

Author Response

Thank you for taking the time to review my manuscript and for providing such positive feedback. I appreciate that your recommendation for publication as is. However, other reviewers suggest several comments to further improve or clarify the manuscript. Now I have revised based on these comments.

Round 2

Reviewer 1 Report

Comments and Suggestions for Authors

The authors outline the historical development of antibody research and key technologies. In their revised version all issues have been adressed adequatly.